# Establishing the Reliability of the GaitON^®^ Motion Analysis System: A Foundational Study for Gait and Posture Analysis in a Healthy Population

**DOI:** 10.3390/s24216884

**Published:** 2024-10-26

**Authors:** Md Farhan Alam, Saima Zaki, Saurabh Sharma, Shibili Nuhmani

**Affiliations:** 1Centre for Physiotherapy and Rehabilitation Sciences, Jamia Millia Islamia, Maulana Muhammad Ali Jauhar Marg, New Delhi 110025, India; farhanphysio@gmail.com (M.F.A.); or saima.zaki@sharda.ac.in (S.Z.); 2Department of Physiotherapy, Sharda School of Allied Health Sciences, Sharda University, Greater Noida 201310, India; 3Department of Physical Therapy, College of Applied Medical Sciences, Imam Abdulrahman Bin Faisal University, Dammam 31441, Saudi Arabia; snuhmani@iau.edu.sa

**Keywords:** 2D motion analysis, kinematics, walking, Auptimo, standard error of measurement, smallest detectable difference

## Abstract

Background: Gait and posture analysis plays a crucial role in understanding human movement, with significant applications in rehabilitation, sports science, and clinical settings. The GaitON^®^ system, a 2D motion analysis tool, provides an accessible and cost-effective method for assessing gait and posture. However, its reliability in clinical practice, particularly for intra-rater consistency, remains to be evaluated. This study aims to assess the intra-rater reliability of the GaitON^®^ system in a healthy population, focusing on gait and posture parameters. Methods: A total of 20 healthy participants (10 males and 10 females) aged 18 to 50 years were recruited for the study. Each participant underwent gait and posture assessments using the GaitON^®^ system on two separate occasions, spaced one week apart. Video recordings from anterior and posterior views were used to analyze gait, while images from anterior, posterior, and lateral views were captured to assess posture with markers placed on key anatomical landmarks. The reliability of the measurements was analyzed using intraclass correlation coefficients (ICC), a standard error of measurement (SEM), and the smallest detectable difference (SDD) method. Results: The GaitON^®^ system demonstrated excellent intra-rater reliability across a wide range of gait and posture parameters. ICC values for gait parameters, including hip, knee, and ankle joint angles, ranged from 0.90 to 0.979, indicating strong consistency in repeated measurements. Similarly, ICC values for posture parameters, such as the head alignment, shoulder position, and ASIS alignment, were above 0.90, reflecting excellent reliability. SEM values were low across all parameters, with the smallest SEM recorded for the hip joint angle (0.37°), and SDD values further confirmed the precision of the system. Conclusion: The GaitON^®^ system provides reliable and consistent measurements for both gait and posture analysis in healthy individuals. Its high intra-rater reliability and low measurement error make it a promising tool for clinical and sports applications. Further research is needed to validate its use in clinical populations and compare its performance to more complex 3D motion analysis systems.

## 1. Introduction

Gait and posture analysis is essential for understanding human locomotion and movement mechanics, playing a critical role in rehabilitation, ergonomics, and sports science [1]. It also serves as a cornerstone in diagnosing and managing a wide range of musculoskeletal [2] and neurological disorders [3]. These assessments enable professionals to monitor performance and recovery, providing a baseline for designing effective treatment or training programs. Advances in human movement research have been significantly driven by the growing demand in medicine and sports for precise methods to capture and refine data on human movement dynamics [4,5]. The assessment of human motion during functional activities is particularly crucial in both rehabilitative medicine and sports science. However, there remains a strong imperative to adapt the application of each system to meet specific contextual needs.

In some instances, a 2D biomechanical analysis, including a posture assessment, offers a rapid and efficient method of evaluation, especially for movements like walking or running that primarily occur in the sagittal plane, thereby avoiding the need for more complex techniques. Conversely, when the movement analysis requires examining multiple planes or investigating forces, a 3D system becomes more appropriate, though it demands a higher level of expertise for an accurate assessment. Quantitative analysis of human movements and postures is a powerful tool for evaluating movement execution, identifying potential injury risk factors [6], aiding clinicians in making informed decisions to reduce patients’ recovery time, and guiding the development of effective treatment plans [7].

Walking patterns and postures exhibit significant variability across different conditions, highlighting the critical importance of comprehensive gait and posture analysis. For instance, elderly individuals often display wider stride widths and reduced walking speeds compared to younger adults, especially when facing proprioceptive challenges [8]. In older adults with diabetic peripheral neuropathy, spatiotemporal gait parameters are notably affected, necessitating targeted sensorimotor training [9]. Additionally, those with an altered foot posture, such as pronated feet, frequently experience lower-back pain, disability, and postural alterations [10].

Assessing gait, particularly walking speeds, using specialized devices provides valuable insights into an adult’s health and functional status [11,12]. In patients with Parkinson’s disease, distinctive kinematic changes such as a slower gait speed and shorter stride length underscore the profound impact of neurological conditions on locomotion [13]. Moreover, slow walking presents a greater challenge to motor control and may be more sensitive to age-related declines in gait compared to walking at normal or faster speeds [14]. Environmental factors, such as the surface type (natural versus artificial) and dual-task scenarios, further influence gait variability and complexity [15]. Research also suggests that postural stability varies depending on whether individuals walk with their eyes open or closed, revealing adaptive strategies employed by the postural control system [16]. Additionally, awkward positions adopted during work can lead to muscular imbalances and postural changes. These compensatory postures often result in musculoskeletal pain and are significant contributors to occupational diseases [17,18,19,20].

Gait and posture alterations are also evident in various other musculoskeletal conditions. Previous studies have reported changes in spatiotemporal parameters in individuals with knee osteoarthritis (OA), demonstrating that these patients walk more slowly, with reduced stride length and lower single-limb support compared to controls [21,22]. Additionally, kinematic alterations during specific phases of the gait cycle in knee OA patients have been observed, including decreased knee excursion during flexion, reduced peak flexion during stance and increased knee flexion at the heel strike [23,24,25]. In individuals with lower-back pain (LBP), gait is characterized by reduced variability in upper body movements, indicating decreased flexibility in trunk coordination. This rigidity is further exacerbated during attention-demanding tasks, suggesting that LBP sufferers exhibit tightened gait control, likely due to a heightened cognitive regulation of gait coordination. However, these alterations in gait coordination diminish the ability to effectively respond to unexpected perturbations, making them maladaptive [26].

Various pieces of equipment and devices are available for the assessment of posture [27,28,29] and gait analysis [30,31,32,33,34,35], with their reliability and validity being periodically evaluated. While 3D imaging is widely used for motion and gait analysis, it can also be applied to assess the relative position of static anatomical landmarks in postural evaluations. However, the utilization of 3D motion analysis systems is limited by several factors, including the high cost of equipment, the need for specialized operator training, the complexity of data processing, and the requirement for a large, non-portable setup [36,37,38,39,40]. These constraints restrict the use of 3D systems, primarily in academic and research settings. As a result, there is a growing need for more cost-effective and accessible methods of posture and gait assessment that health professionals can easily utilize in clinical practice beyond the confines of academic and research settings.

Two-dimensional gait and posture analysis systems are more cost-effective and portable compared to 3D systems and are simpler to set up and use, making them accessible for clinicians with limited resources or expertise [30,41,42]. Despite their relative simplicity, recent studies have shown that 2D systems can produce valid data, as supported by the existing literature [31,38,41,42]. Additionally, the capability for slow-motion and frame-by-frame analysis allows for the observation of subtle movement details, which is crucial for identifying early signs of injury or biomechanical inefficiencies.

Previously, the reliability of 2D motion analysis systems has been widely evaluated across various applications. The Kinovea system, for instance, has demonstrated reliability in assessing left lower limb kinematics during the initial contact phase in the sagittal plane [41] and spatiotemporal gait parameters [42]. Additionally, it has been evaluated for the knee range of motion (ROM) in a supine position [43] and for trunk forward tilts during gait [44]. Other 2D motion analysis systems, such as the Stride Analyzer for knee osteoarthritis [45] and GaitMat II in healthy subjects [46], have shown reliability for spatiotemporal parameters but did not assess kinematic data. In contrast, the Quintic Biomechanics software package (9.03 version 17, Quintic Consultancy Ltd., Coventry, UK) has been validated for measuring the Frontal Plane Projection Angle (FPPA) during single limb squats [47] and for tibial angles in elite football players [48], demonstrating its broader applicability in sports settings.

Previous studies on the reliability of 2D posture analysis often lack a comprehensive assessment of posture across all perspectives, including anterior, posterior, and lateral views. For instance, Greisberger et al. (2019) assessed the reliability of Templo 7.1 2D Motion Analysis Software (Contemplas, Kempten, Germany) for measuring the Craniovertebral Angle (CVA) and Trunk Forward Lean (TFL) [49]. More recently, Carrasco-Uribarren et al. (2023) evaluated a smartphone-based computer vision application for CVA, focusing on test–retest and inter-rater reliability, as well as concurrent validity, in both healthy individuals and those with neck pain or tension-type headaches [50]. These studies show the potential of 2D systems for specific metrics but highlight the need for broader evaluations across multiple views.

Among these, GaitON^®^ stands out as a portable and user-friendly 2D motion analysis system developed by Auptimo Technologies LLP (Delhi, India) in 2017. Specifically designed to streamline biomechanical assessments for healthcare professionals and movement specialists, GaitON^®^ offers an accessible, efficient solution for gait and posture analysis in clinical and sports environments, making high-quality motion analysis more approachable for a broader audience.

The GaitON^®^ system is equipped with a range of versatile features, making it an ideal tool for both clinical and research applications. It offers comprehensive assessment modules, including static assessments (e.g., standing and sitting posture) and dynamic capabilities (e.g., walking and running gait, tennis serve, golf swing, and cycling analysis). Its flexible video input supports analysis using various camera types, from mobile phones to high-end digital cameras, ensuring adaptability to diverse research setups. GaitON^®^ also includes efficient data processing modules, significantly reducing analysis time, and generates structured reports with normative values for easy interpretation. Furthermore, it provides all calculated data and reference values in Excel format, simplifying data organization and further analysis for researchers. Additionally, beyond its built-in analysis capabilities, the system allows for precise manual measurements of angles, such as pelvic tilts and lumbar flexibility. It also offers detailed spatiotemporal parameter analysis, providing a comprehensive understanding of gait mechanics.

However, ensuring the reliability of this 2D motion analysis system for assessing gait and posture, particularly in terms of intra-rater reliability, is crucial for its application in clinical settings to provide valid assessments and outcomes. To date, no study has been conducted to evaluate the intra-rater reliability of the GaitON^®^ system. Therefore, the current study aims to assess the intra-rater reliability of the GaitON^®^ system for analyzing kinematic parameters of gait and standing posture. Additionally, it seeks to examine the standard error of measurement (SEM) and smallest detectable difference (SDD) in these assessments.

## 2. Methods

### 2.1. Study Design

This study aims to evaluate the reliability of the GaitOn^®^ software (V1.9, Auptimo Technologies LLP, Delhi, India) for assessing gait and standing posture in healthy individuals. It forms part of a broader trial registered with the Clinical Trials Registry—India (CTRI)—under the identifier CTRI/2024/01/061348. The research was conducted in compliance with the ethical principles outlined in the 1964 Declaration of Helsinki and received approval from the Institutional Human Ethics Committee at Jamia Millia Islamia, New Delhi, India (approval number 15/2/439/JMI/IEC/2023).

### 2.2. Participants

Twenty young, healthy individuals (10 males and 10 females), aged 18 to 50 years, with a body mass index (BMI) ranging from 18 to 25 kg/m^2^, voluntarily participated in the study. The demographic characteristics of the participants are detailed in Table 1. Individuals with musculoskeletal, neurological, or other medical conditions that could impair balance or hinder their ability to perform gait tasks were excluded from the study. Prior to participation, all participants were thoroughly informed about the study and provided written informed consent.

### 2.3. Experimental Procedure

#### 2.3.1. Participant Preparation

All participants were provided with a thorough explanation of the 2D motion analysis system and the entire procedure that would be conducted. They were given specific instructions regarding appropriate clothing for the video capture process, ensuring areas were either exposed or fitted tightly to facilitate accurate marker placement on both the upper and lower body. The suitability of each participant’s clothing was carefully verified prior to the commencement of the video capture process. Participants were advised to wear specially designed attire for gait and posture analysis, as recommended by the GaitON team (Figure 1a,b). Self-adhesive green/pink fluorescent markers with a diameter of 12 mm or ball markers were applied to all participants at the designated anatomical landmarks for gait and posture analysis (Figure 2a,b)

#### 2.3.2. Marker Placement

Gait analysis: To conduct gait analysis using the GaitON^®^ software (V1.9, Auptimo Technologies LLP, Delhi, India), patients were evaluated from two positions: posterior and lateral views. In the lateral view, markers were positioned at the greater trochanter of the femur, the lateral epicondyle of the femur, and the lateral malleolus. Additionally, two markers were placed parallel to the sole of the foot—one below the lateral malleolus and the other at the head of the fifth metatarsal (Figure 3a). All markers were placed bilaterally on both lower limbs. For the posterior view, markers were strategically placed on the posterior aspect of the leg and foot. The first marker was positioned at the base of the calcaneus, while the second marker was placed at the Achilles tendon attachment. The third marker was located at the center of the Achilles tendon at the level of the medial malleolus, and the fourth marker was placed 15 cm above the third marker, centered on the calf (Figure 3b). These markers were applied bilaterally on both the right and left sides. Additionally, two markers were placed on the posterior superior iliac spine (PSIS) on both sides (Figure 3c) (Appendix A).

Posture analysis: To perform postural analysis using the GaitON^®^ software posture analysis system, patients were assessed from three positions: anterior, posterior, and lateral views. For the anterior view, markers were strategically placed at three levels on both the right and left sides—specifically at the level of the antero-superior iliac spine (ASIS), the center of the patella, and the tibial tuberosity (Figure 3d). Additionally, two markers were placed bilaterally at the tip of the acromion, ensuring comprehensive alignment assessment (Figure 3f).

For the posterior view, the first marker was placed at the base of the heel (calcaneus), and the second at the insertion of the Achilles tendon, establishing one axis. The third marker was positioned at the center of the Achilles tendon at the level of the medial malleolus, while the fourth marker was placed 15 cm above the third marker, centered on the calf, forming the second axis (Figure 3b). These markers were applied symmetrically on both the right and left sides. For the lateral views, markers were placed at the following anatomical landmarks: the C7 spinous process (Figure 3e), the midpoint of the humeral head (Figure 3f), the greater trochanter of the femur, the lateral femoral epicondyle, and the lateral malleolus (Figure 3a). These markers were positioned on both the right and left sides (Appendix A).

#### 2.3.3. Setting Up of 2D Gait Laboratory

A linear walkway of approximately 6 m, with non-slip flooring and adequate lighting, was selected for the gait assessment. Two high-speed video cameras (Logitech, resolution 1920 × 1080 p at 30 frames per second (fps), shutter speed 1/250 s, diagonal field view 90°) were mounted on professional video recording tripods, ideally in landscape mode, at a height of 0.63 m (Figure 4). One camera was positioned anteriorly to capture both anterior and posterior views, while the other was placed laterally to capture the lateral view. These cameras were connected to a laptop via USB cables to capture video recordings and images. The optimal lateral tripod placement was determined to be approximately 2.5 m, ensuring that the subject’s appropriate frame remained within the camera’s field of view for gait analysis and allowing for the completion of at least one full gait cycle within the captured frame. The complete setup of the gait lab is illustrated in the accompanying figure, providing a detailed overview of the experimental environment (Figure 5a).

The camera tripods were equipped with bubble levels, which were centered to level the cameras along their pitch and roll axes, ensuring that the cameras were perpendicular to the ground and minimizing parallax errors. To further ensure accuracy, the camera optical axes were aligned perpendicular to the plane of motion using a grid calibration method. An axis calibration frame of size 11 inch × 23 inch with two perpendicular grid lines was placed in the middle of the walkway to verify the proper alignment of the anterior and lateral axes, as illustrated in the accompanying figure (Figure 5b). The alignment of the grid lines was checked on a laptop (Figure 5c), and once confirmed, the mat was removed to allow for unobstructed gait recording.

#### 2.3.4. Posture Assessment Setup

In the posture assessment protocol utilizing the GaitON^®^, a meticulously structured environment was established to ensure the accuracy and reliability of data collection. The central component of this setup was a calibrated wall grid (height: 213.36 cm, width: 99.44 cm), which served as a reference for aligning and measuring postural deviations. This grid provided a consistent and standardized background against which the patient’s posture was evaluated, ensuring precise alignment in the anterior, posterior, and lateral views. A single high-resolution camera (Logitech, resolution 1920 × 1080 p, diagonal field view 90°) was mounted on a professional video recording tripod equipped with bubble levels and was positioned to capture the subject’s full body within the frame, ensuring accurate recording of all necessary postural metrics. The camera was positioned at a height of 135 cm and placed at an approximate distance of 250 cm from the calibrated wall grid to ensure optimal capture conditions. The full setup of the posture lab is depicted in the accompanying figure (Figure 6), offering a comprehensive view of the experimental setup and environment.

#### 2.3.5. Procedure for Gait and Posture Recording

All participants were prepared for gait analysis by wearing appropriate attire as instructed and having markers applied to the relevant anatomical landmarks. Prior to the actual recordings, participants were instructed to walk barefoot on the analysis surface as a warm-up session to acclimate themselves to the testing environment. To eliminate any potential learning effects, all participants were familiarized with the testing procedures beforehand. Participants were directed to walk at their normal, comfortable pace. Video recordings were conducted for each participant for a duration of 30 to 60 s in both the anteroposterior and lateral views, ensuring sufficient time to capture more than two rounds in the walking area.

Posture analysis began with the application of markers to the relevant anatomical landmarks, after which subjects were instructed to march first in place before assuming the final standing position. They were then asked to stand in anterior, posterior, and lateral views, one at a time, with their posture recorded against the wall grid. For both gait and posture assessments, each participant was recorded on two separate occasions, with an approximately one-week interval between the sessions.

### 2.4. Gait and Posture Outcomes

For gait assessment, the anterior, posterior, and lateral views are typically utilized. However, in the current study, only the lateral and posterior views were employed. The lateral view was particularly focused on analyzing the phases of the gait cycle, with kinematic data being recorded for the angles of the ankle, knee, and hip joints during each phase. Additionally, the posterior view was used to measure the rear foot angle for the right limb and pelvic drop. For the posture assessment, the anterior, posterior, and right lateral views were utilized. From the anterior view, the horizontal alignment of the head, acromions, and anterior superior iliac spines (ASIS), as well as the Q angle, were recorded. The posterior view was used to calculate the rear foot angle (calcaneal eversion) of the right limb, while the right lateral view provided measurements of the forward head angle, shoulder angle, and genu recurvatum. These outcomes were critical in assessing the reliability of the posture and gait evaluations in this study.

### 2.5. Precision Angle Measurement and Video Analysis

The system processes motion data through advanced image processing techniques. During the analysis, the user manually digitized markers placed on key anatomical landmarks at specific gait cycle events or postural views. Based on the pixel coordinates of these digitized markers, the GaitON^®^ software automatically calculates all relevant angles for gait and posture analysis. The computed data is then exported into comprehensive reports, enabling efficient and accurate interpretation of kinematic parameters.

In this reliability study, angles in the sagittal plane were measured and quantified using a standardized protocol, which has been demonstrated to provide reliable results in previous research [51]. The use of 2D software for measuring these angles has been validated for its sufficient reliability. The hip flexion/extension angle was quantified as the angle between a vertical line perpendicular to the ground and a line connecting the markers placed on the greater trochanter of the femur and the lateral condyle of the femur [52]. Similarly, knee flexion/extension was measured as the angle between the line connecting the markers on the greater trochanter of the femur and the lateral condyle of the femur and the line connecting the markers on the lateral condyle of the femur and the lateral malleolus [52]. Ankle dorsiflexion and plantar flexion were assessed by measuring the angle between the line connecting the markers on the lateral condyle of the femur and the lateral malleolus and the line parallel to the sole of the foot, defined by markers below the lateral malleolus and the 5th metatarsal (Figure 7a).

The pelvic drop was evaluated by drawing a line connecting the markers on both posterior superior iliac spines (PSIS) and assessing the deviation of this line from a horizontal reference line drawn parallel to the ground. The rear foot angle, indicating calcaneal eversion, was calculated as the angle between the line connecting the markers at the base of the heel and the insertion of the Achilles tendon, and the line connecting the marker at the center of the Achilles tendon with a marker placed 15 cm above it (Figure 7b). The Q angle was determined as the angle between the line connecting the markers at the anterior superior iliac spine (ASIS) and the center of the patella and the line connecting the markers at the center of the patella and the tibial tuberosity (Figure 7c).

For horizontal alignment assessments, the alignment of the head was measured by connecting lobulus auriculae of the ears, while acromion alignment was evaluated by drawing a line connecting the tips of the acromions bilaterally. The horizontal alignment of the anterior superior iliac spines (ASIS) was assessed by connecting the markers on the ASIS. These lines were then compared to a horizontal reference line to calculate the actual angles, ensuring precise and reliable measurements in the evaluation of posture (Figure 7c). In the right lateral view for posture assessment, forward head posture was evaluated by measuring the angle formed between a line drawn from the C7 vertebra to the tragus of the ear and a horizontal reference line. The shoulder angle was determined by measuring the angle between a line connecting the C7 vertebra to the mid-point of the humeral head and a horizontal line. For the assessment of genu recurvatum in the right lower limb, the angle was measured between two lines: one connecting the markers on the greater trochanter of the femur and the lateral condyle of the femur, and the other connecting the lateral condyle of the femur to the lateral malleolus (Figure 7d).

Before quantification, a single physiotherapist (rater) independently reviewed each video in two separate sessions, excluding any footage with poor lighting, blurriness, or other quality issues that could potentially bias the digitalization process and affect the accurate quantification of joint angles [53]. Gait and standing posture outcomes were then assessed at specific video frames using the GaitON^®^ software, ensuring precise and reliable measurements.

### 2.6. Rater

In this study, a single rater with two years of experience using the GaitON motion analysis system was responsible for the preparation and execution of all gait and posture assessments. The rater independently set up the GaitON lab, placed the markers, and performed both the measurement and analysis. Prior to the study, the rater completed multiple supervised training sessions under the guidance of an expert with 7 years of experience in the GaitON motion analysis system to ensure proficiency and accuracy in system use. During the study, the rater received further guidance and support from the expert when difficulties or uncertainties arose, ensuring the reliability and consistency of the data collected.

### 2.7. Statistical Analyses

The statistical analysis was conducted using IBM SPSS Statistics version 27 (IBM SPSS, Chicago, IL, USA). The gait and posture variables were measured for each participant on two occasions, with an approximately one-week interval between the sessions. The intra-rater reliability of the test–retest measurements was evaluated using the intraclass correlation coefficient (ICC). The ICC values were interpreted according to the following scale: Poor = <0.40, Fair = 0.40–0.70, Good = 0.70–0.90, and Excellent = >0.90. The SEM was calculated using the formula SD (pooled) × √ (1 − ICC), while the SDD was computed as 1.96 × √2 × SEM to assess the response stability and precision of the measurements.

## 3. Results

The kinematic gait parameters of the right lower limb were analyzed from both the lateral and posterior views. The summary of the mean and standard deviation (SD) of these parameters, along with their reference values, is presented in Table 2. In addition, Table 3 presents the mean and SD of the standing posture parameters captured from the anterior, posterior, and right lateral views, along with their reference values. The ICC for the right hip, knee, and ankle angles demonstrated excellent reliability across different phases of the gait cycle. Specifically, the ICC values for the hip angles ranged from 0.903 at initial contact to 0.979 during the preswing. Similarly, the knee angles with ICC values ranged from 0.902 during terminal stance to 0.972 during the preswing. The ankle angles also showed strong reliability, with ICC values from 0.90 during the initial swing to 0.952 in the preswing phase. Additionally, the ICC value for the rear foot angle and pelvic drop during the mid-stance phase was recorded as 0.904 and 0.929, respectively. The corresponding SEM and SDD values for gait parameters are detailed in Table 4. As shown in Table 5, the ICC, SEM, and SDD for the posture parameters reveal a high level of measurement precision. The ICC values, ranging from 0.902 to 0.945, indicate excellent reliability for posture.

For the gait parameters, the SEM values demonstrated a range of variability across the joints. The SEM values for the hip joint angle ranged from 0.37 to 1.02, while for the knee joint angle, the range was between 0.49 and 0.87. For the ankle joint, SEM values were slightly higher, ranging from 0.87 to 1.29. In terms of SDD, the values for the hip joint ranged from 1.02 to 2.82, while the knee joint values ranged from 1.36 to 2.41. The ankle joint showed the highest SDD range, from 2.40 to 3.57. For the rear foot angle and pelvic drop, the SEM values were 0.53 and 0.41, respectively, with corresponding SDD values of 1.46 for the rear foot angle and 1.12 for the pelvic drop. The SEM for posture parameters varied between 0.15 and 0.91, while the SDD ranged from 0.40 to 2.52, indicating a high degree of precision in posture assessment. Table 6 provides a glossary detailing the gait and posture angles, offering a clear understanding of the joint positions and angles used in the analysis.

## 4. Discussion

The GaitOn^®^ system, a 2D motion analysis tool designed for gait and posture assessment, offers a portable and cost-effective solution with promising potential for clinical use. One of its key advantages is its flexibility, as it allows for the analysis of videos captured from a wide range of devices, including mobile phones and high-end digital cameras. The system’s inbuilt modules for gait and posture significantly reduce analysis time, streamlining the evaluation process. Additionally, GaitON^®^ generates comprehensive reports that include normative values for each activity, facilitating quick and accurate data interpretation and making it an efficient tool for clinical use. However, given that the system captures motion in only two dimensions, there is a potential risk of parallax and perspective errors during measurement. As a result, it is essential to establish the system’s reliability and measurement accuracy before it can be integrated into clinical decision-making processes. Studies specifically addressing the reliability of the GaitOn^®^ system for gait and standing posture assessments are still lacking. This study aims to address that gap by evaluating the reliability of the GaitOn^®^ system for these specific purposes.

In this investigation, the GaitOn^®^ software was used to evaluate its reliability in analyzing gait and posture in a healthy population. To our knowledge, this is the first study focused on assessing the reliability of the GaitOn^®^ system for gait and posture analysis in healthy individuals. The results provide robust preliminary evidence of the system’s excellent intra-rater reliability. The ICC values were consistently high across different phases of the gait cycle and posture assessments, underscoring the GaitOn^®^ system’s reliability for capturing and analyzing lower limb kinematics and postural parameters. These findings confirm the system’s capability to deliver consistent and reproducible measurements in healthy individuals.

Additionally, this study presents the SEM and SDD values for both gait and posture parameters. The SEM provides critical insights for clinicians and patients by helping gauge the effectiveness of rehabilitation protocols, indicating when therapeutic goals are being achieved. In contrast, the SDD reflects significant changes in performance, allowing for the identification of measurable improvements or deteriorations in the patient’s condition. This provides a more meaningful assessment of progress and underscores the practical value of the GaitOn^®^ system in clinical settings.

The results of the study demonstrate excellent intra-rater reliability for gait parameters measured using the GaitON^®^ system, as reflected by high ICC values across all joints and phases of the gait cycle. The ICC values for ankle, knee, and hip angles range from 0.90 to 0.979, indicating strong consistency in repeated measurements, especially during critical phases such as preswing and terminal stance. Additionally, the low SEM values, such as 0.37° for the hip angle in preswing and 0.49° for the knee angle during the initial contact, suggest minimal measurement error. The SDD values further support the system’s precision, with small detectable differences across parameters, such as 1.02° for the hip angle in preswing and 1.36° for the knee angle at initial contact. These findings highlight the GaitON^®^ system’s ability to detect even subtle changes in gait parameters.

In addition to the high reliability of the GaitON^®^ system, the variability in SEM and SDD values between different phases of the gait cycle is noteworthy. For example, the ankle joint, which exhibited ICC values ranging from 0.90 to 0.952, had higher SEM values, particularly during the preswing phase (1.29°), compared to the terminal stance phase (0.87°). This suggests that while the system provides consistent measurements, there is slightly more measurement error in phases with more dynamic movement, such as preswing. Similarly, the hip joint showed a highly reliable ICC of 0.979 during the preswing, with an impressively low SEM of 0.37°, indicating exceptional precision in measuring hip movement during this critical phase. The rear foot angle and pelvic drop also demonstrated excellent reliability with ICC values of 0.904 and 0.929, respectively, and small SEM values, reinforcing the system’s capability to provide precise measurements even for more complex parameters like pelvic drop, where the SEM was only 0.41°. These results further solidify the GaitON^®^ system’s effectiveness in capturing a wide range of gait parameters with minimal error.

The posture parameters assessed using the GaitON^®^ system demonstrated excellent intra-rater reliability, as reflected by high ICC values across various views. The anterior view parameters, such as the horizontal alignment of the head, acromions, and ASIS, exhibited ICC values above 0.90, with minimal SEM and SDD values, indicating strong measurement precision. For instance, the horizontal alignment of the head had an ICC of 0.925 and an SEM of 0.15°, suggesting minimal variability in repeated measures. Similarly, the rear foot angle from the posterior view demonstrated strong reliability (ICC = 0.947) with a low SEM of 0.34°, underscoring the GaitON^®^ system’s ability to capture even subtle postural deviations with high accuracy. Parameters such as forward head angle and shoulder angle in the right lateral view also showed high reliability, although the SEM and SDD values were slightly higher compared to other parameters, particularly for shoulder angle (SEM = 0.91°, SDD = 2.52°).

Numerous studies have investigated the reliability of 3D motion analysis systems [54,55,56,57,58] and 2D motion analysis systems [41,59] in assessing human gait across diverse populations. These systems have proven to be valuable tools for precise movement evaluation. Notably, 2D motion analysis systems have also demonstrated reliability in the assessment of posture [44,60] and running mechanics [53,61,62], further expanding their applications in both clinical and performance settings.

A previous study assessing the reliability of Kinovea^®^ software (version 0.8.15) for gait analysis reported high intra-rater reliability for hip, knee, and ankle angles during the initial contact phase of gait, with ICCs exceeding 0.85 for all joints. In comparison, our results showed even greater reliability, with ICC values exceeding 0.90 for each phase of the gait cycle. However, discrepancies between Kinovea^®^ and the Vicon® 3D Motion Analysis System- Vicon Nexus® 1.8.5 software (Oxford Metrics, Oxford, UK) were observed, with Bland–Altman plots indicating disagreements of ±2.5° to ±5° across joints. These differences suggest caution in interpreting small joint angle variations, as they may impact clinical decision-making [41]. Furthermore, another study demonstrated that a 2D video system reliably assessed adaptive gait kinematics in both healthy individuals and those with central vision loss. The system showed strong agreement with the gold standard 3D Vicon system, with Pearson’s correlations ranging from 0.755 to 0.997 for key gait parameters such as the crossing height, foot placement, and crossing velocity. Test–retest reliability was also high, with ICC values exceeding 0.93. Although minor systematic biases were noted in specific measurements, corrections brought the 2D system’s performance closer to the 3D standard [59].

In terms of postural assessment reliability, two-dimensional (2D) motion analysis systems have demonstrated excellent consistency in various studies. The Kinovea system demonstrated high reliability in measuring forward trunk tilt during gait, with ICCs exceeding 0.92 for both maximum and minimum tilt angles. These findings align closely with our results, as we also observed ICC values above 0.90 in the assessment of standing posture. The system also exhibited low SEM and the smallest detectable difference (MDC95), indicating its precision in clinical settings with minimal margin for error compared to more complex 3D systems [44]. In comparison to the findings from the Kinovea system, where the SEM ranged from 0.98 to 1.06, and the smallest detectable difference (MDC95) was between 2.7° and 2.9° for a forward trunk tilt during gait [44], our study demonstrated slightly better precision for standing posture assessments. Specifically, our SEM value was 0.83, which is lower than that reported in the Kinovea study, indicating greater accuracy in our measurements. Additionally, the SDD in our study was 2.23°, which is also lower than the MDC95 range reported for the Kinovea system, suggesting that our postural assessment system can detect smaller changes in posture with higher sensitivity. These comparisons highlight the reliability and precision of our posture assessment in standing positions.

Similarly, another study assessing 2D motion analysis for quantifying postural deficits in adults, including those with neurological impairments, reported excellent reliability across a range of postural tasks. ICC values ranged from 0.89 to 1.0 for test–retest reliability and 0.77 to 1.0 for inter-rater reliability, further supporting the utility of 2D systems for accurate postural assessments. These findings reinforce the value of low-cost 2D motion analysis tools, such as Kinovea and GaitON^®^, as viable alternatives to 3D systems for clinical gait and posture analysis, particularly in resource-limited environments where accessibility and cost are major considerations [60].

### 4.1. Strength

The results of this study provide a valuable benchmark for understanding the application of the GaitON^®^ system in gait and posture analysis. While our research focused on a healthy population, these findings establish a foundational reference point for future studies that can explore the system’s use in patients with musculoskeletal conditions. This opens the door for further research to assess the broader clinical applicability of the GaitON^®^ system in clinical and rehabilitation settings.

Building on this foundation, the primary strength of this study lies in its robust methodology, characterized by the use of standardized protocols for marker placement, video recording, and data analysis, as well as the system’s capability to auto-generate analysis. This approach ensures consistency across all participants and minimizes variability, providing a controlled environment that is particularly effective for assessing the reliability of the GaitON^®^ system. Such a controlled design forms a solid basis for future research involving clinical populations or those with musculoskeletal and neurological disorders. The study also benefits from a comprehensive assessment of both gait and posture parameters, covering a wide range of biomechanical metrics, including joint angles and postural alignment across multiple views and all events of the gait cycle (e.g., initial contact, loading response, mid-stance, terminal stance, preswing, initial swing, mid-swing, and terminal swing). This detailed analysis not only enhances the understanding of the system’s versatility but also highlights its potential to assess both dynamic and static aspects of human movement. Additionally, the system’s versatility in field-based applications, especially in sports performance analysis, along with its ability to generate standardized reports, makes it a valuable tool for comprehensive evaluation across diverse clinical and research settings.

Another major strength is the study’s focus on intra-rater reliability, ensuring that the system provides consistent measurements by the same rater over time. The use of ICC, SEM, and SDD offers a detailed examination of the system’s precision, demonstrating its ability to detect subtle changes in movement. The accessibility and cost-effectiveness of the GaitON^®^ system add further value, as it presents a viable alternative to more complex and expensive 3D motion analysis systems, particularly in resource-limited clinical settings. Moreover, the statistical rigor applied in this study, with well-established metrics such as ICC, SEM, and SDD, ensures that the findings are both statistically robust and clinically relevant, supporting the potential integration of the GaitON^®^ system into routine clinical practice and broader research applications.

### 4.2. Limitations

Despite the strengths of this study, there are several limitations that should be acknowledged. First, the sample size was relatively small, with only 20 healthy participants. While this number is sufficient to establish baseline reliability, larger and more diverse populations, including individuals with musculoskeletal or neurological disorders, are necessary to fully assess the system’s generalizability and clinical applicability. The study also focused solely on intra-rater reliability, meaning that the consistency of measurements across different raters was not evaluated. Inter-rater reliability is essential for broader clinical use, where multiple practitioners may be involved in patient assessment. Without this, the findings may not fully reflect the system’s reliability in real-world settings where multiple users are involved.

Another limitation is the reliance on 2D motion analysis, which inherently carries the risk of parallax and perspective errors, particularly when analyzing movements that occur in multiple planes. Being a 2D motion analysis system, GaitON is limited to data parameters in the frontal and sagittal plane and is unable to process transverse plane data compared to more advanced 3D systems. Additionally, while the study demonstrated the system’s precision for static and dynamic assessments, the results may not fully capture the variability in movement patterns seen in clinical populations with impaired gait or posture. Therefore, further studies are needed to validate the GaitON^®^ system in these more complex scenarios, as well as to compare its performance directly with 3D motion analysis systems for a more comprehensive understanding of its strengths and limitations.

### 4.3. Future Perspectives

Looking ahead, future research should focus on expanding the application of the GaitON^®^ system to more diverse populations, particularly those with musculoskeletal, neurological, or postural disorders. Evaluating the system’s reliability in clinical settings where movement impairments are present will be crucial to determine its utility in diagnosing and monitoring rehabilitation progress. Furthermore, studies involving larger sample sizes and multiple raters will help establish both intra- and inter-rater reliability, enhancing the system’s robustness and ensuring consistent outcomes across different clinical practitioners. In addition, the system could be tested in dynamic scenarios, such as during sports activities or in elderly populations where gait and balance are more variable, to assess its full potential in a variety of real-world applications.

Technological advancements could also offer significant improvements to the GaitON^®^ system. Integrating additional sensors or combining 2D motion capture with 3D analysis could provide a more comprehensive view of human movement while maintaining the system’s accessibility and cost-efficiency. Furthermore, the development of machine learning algorithms for automated marker placement and data analysis could reduce user dependency and increase the system’s accuracy and efficiency. Such innovations would not only improve the reliability of measurements but also open new avenues for use in telemedicine, where remote monitoring of gait and posture could support early diagnosis and intervention.

### 4.4. Clinical Implications

The findings of this study have significant clinical implications, particularly in the fields of rehabilitation, sports science, and general clinical practice. The GaitON^®^ system, with its demonstrated intra-rater reliability, provides clinicians with a reliable, cost-effective tool for assessing gait and posture in healthy individuals. Its portability and ease of use make it an attractive option for routine clinical assessments, particularly in settings where access to more complex and expensive 3D motion analysis systems is limited. This can facilitate more widespread adoption of biomechanical assessments in everyday clinical practice, allowing for better monitoring of patient progress, identification of movement impairments, and the development of individualized treatment plans.

Furthermore, the system’s ability to provide accurate measurements of key kinematic and postural parameters can aid in early diagnosis and intervention for patients at risk of developing musculoskeletal issues or those undergoing rehabilitation. Clinicians can use the GaitON^®^ system to track improvements in gait or posture following interventions, ensuring that treatment is both effective and appropriately adjusted as the patient progresses.

## 5. Conclusions

In conclusion, this study demonstrated the excellent intra-rater reliability of the GaitON^®^ system for assessing both gait and standing posture in a healthy population. The system achieved high ICCs for key kinematic parameters, including hip, knee, and ankle joint angles, confirming its precision in repeated measurements across various phases of the gait cycle, such as preswing and terminal stance. The low SEM values, such as 0.37° for the hip angle in preswing, and the small SDD values further support the system’s ability to detect subtle changes in movement. These findings highlight the GaitON^®^ system as a valuable tool for tracking biomechanical alterations over time, offering reliable data that can capture even minor progressions in gait and posture.

Moreover, the GaitON^®^ system proved to be an accessible and efficient alternative to more complex and expensive 3D motion analysis systems. Its portability and flexibility—accommodating various camera types—make it highly suitable for diverse clinical settings, including resource-limited environments and field-based applications in sports performance analysis. The in-built modules for gait and posture assessments significantly reduce the analysis time, while the system’s ability to generate standardized reports with normative data enhances its clinical utility, allowing for the quick and accurate interpretation of kinematic data. While this study provides a strong foundation for using the GaitON^®^ system in clinical practice and research with healthy populations, future studies should explore its applicability in clinical populations with musculoskeletal and neurological conditions. Additionally, investigating its integration into telemedicine and home-based rehabilitation could expand its potential, making it a versatile solution for remote monitoring and early intervention.

## Figures and Tables

**Figure 1 sensors-24-06884-f001:**
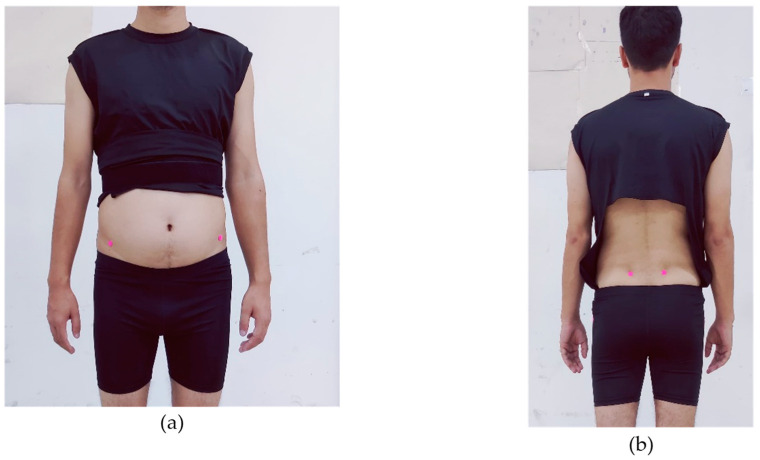
Participant’s attire during assessment using the GaitON^®^ motion analysis system. (**a**) Anterior view, (**b**) posterior view.

**Figure 2 sensors-24-06884-f002:**
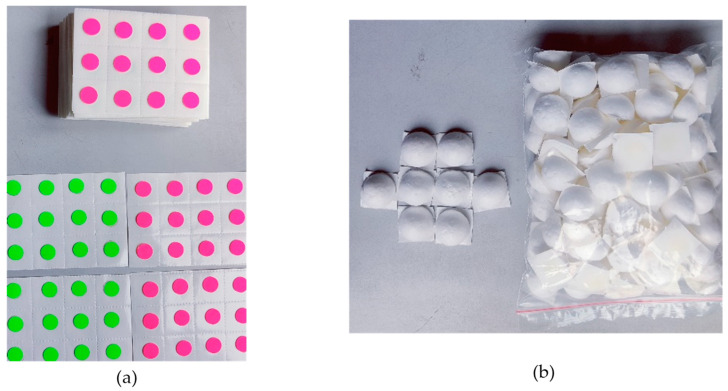
Types of markers used for gait and posture assessment. (**a**) Self-adhesive green/pink fluorescent markers, (**b**) ball markers.

**Figure 3 sensors-24-06884-f003:**
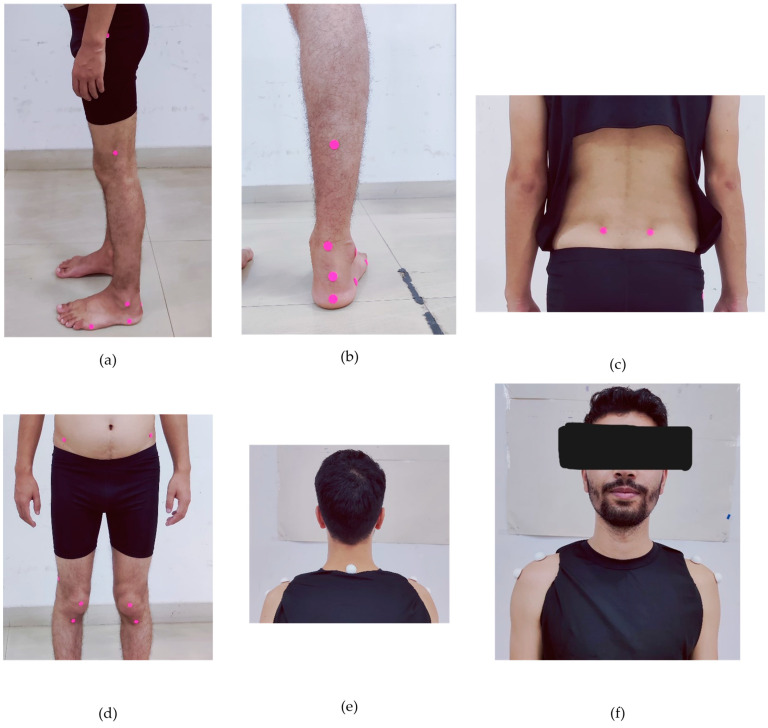
Marker placements for gait and posture assessment using the GaitON^®^ motion analysis system. (**a**) Lateral markers on the lower limbs, (**b**) posterior markers on the posterior leg and foot, (**c**) marker on the posterior superior iliac spine (PSIS), (**d**) anterior markers on the anterior superior iliac spine (ASIS), patella, and tibial tuberosity, (**e**) C7 vertebra marker, and (**f**) markers on the humeral head and acromion.

**Figure 4 sensors-24-06884-f004:**
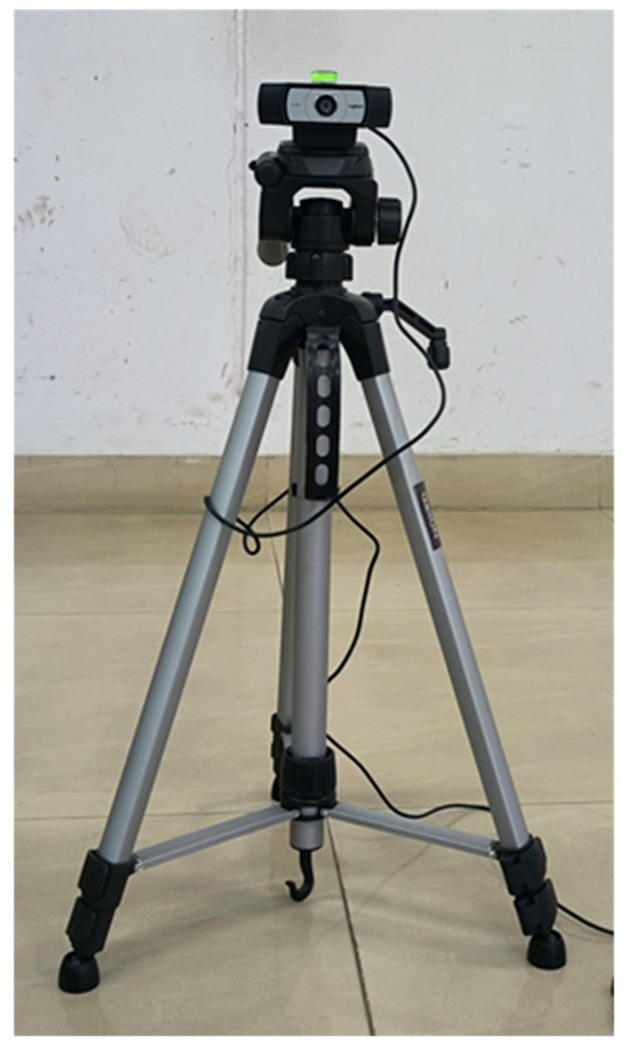
Tripods with cameras setup for the gait and posture analysis laboratory.

**Figure 5 sensors-24-06884-f005:**
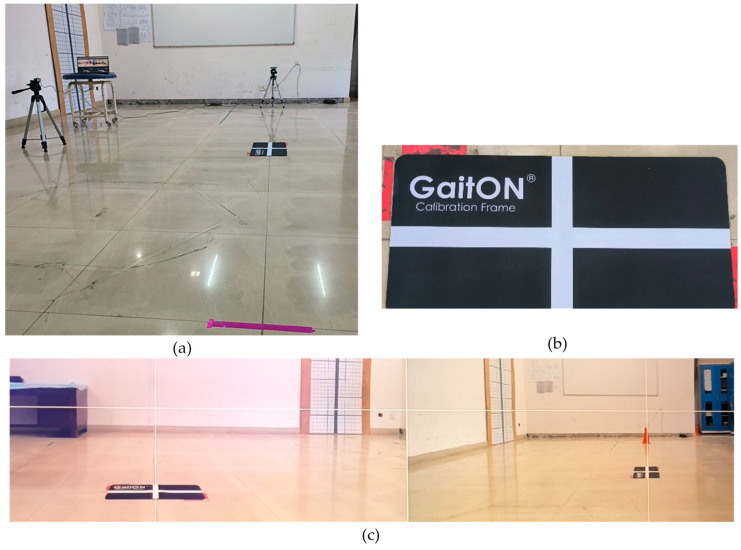
Setup and calibration process of the gait laboratory used for motion analysis with the GaitON system. (**a**) Complete setup of the gait lab with cameras and walkway, (**b**) GaitON Calibration Frame used for axis alignment, (**c**) Alignment of the anterior and lateral axes using calibration tools for accurate motion capture.

**Figure 6 sensors-24-06884-f006:**
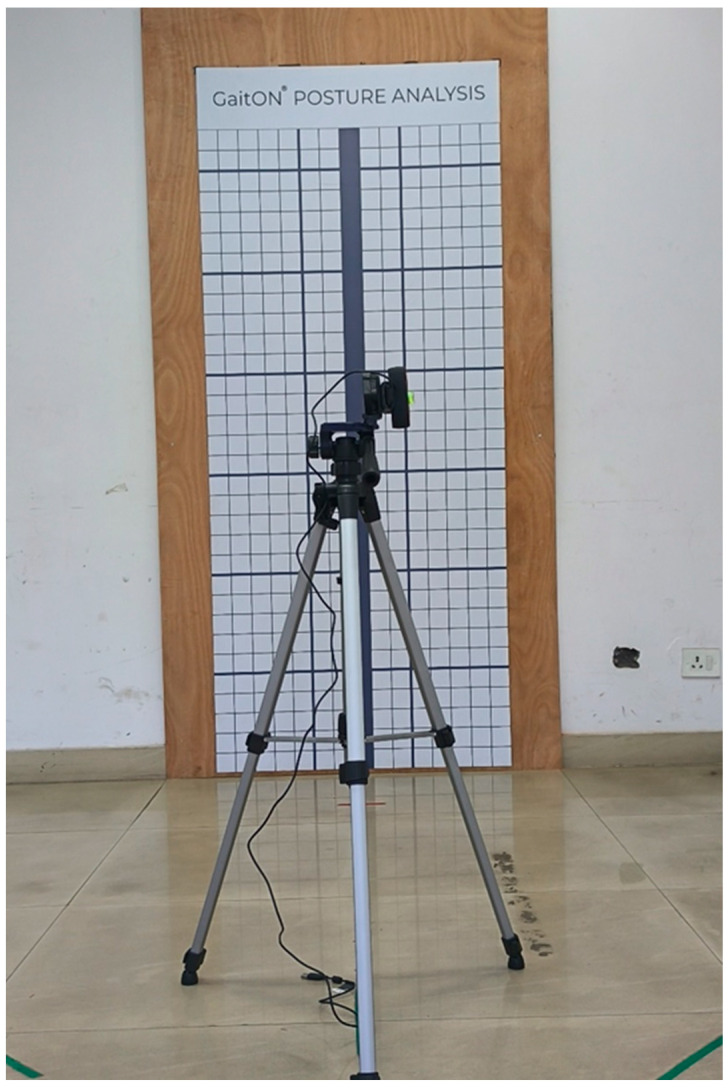
Posture assessment setup.

**Figure 7 sensors-24-06884-f007:**
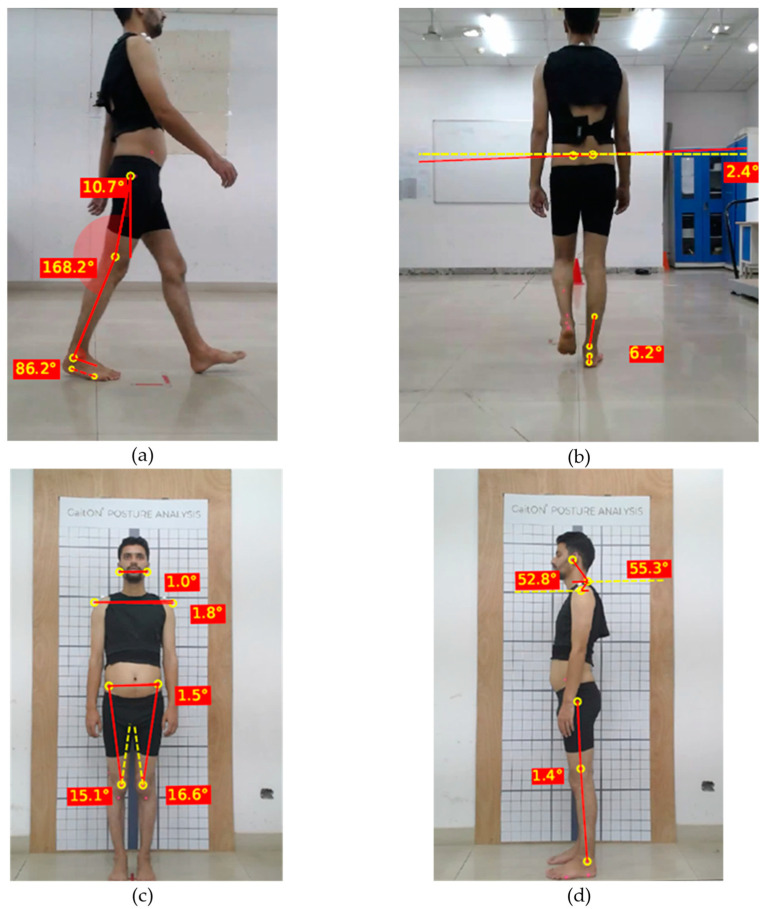
Angle measurement: (**a**) Gait: lateral view, (**b**) Gait: posterior view, (**c**) Posture: anterior view, (**d**) Posture: right lateral view.

**Table 1 sensors-24-06884-t001:** Descriptive statistics (demographics).

Type	Mean ± SD (95% CI)	Range
Age (years)	31.45 ± 7.60 (27.89–35.01)	21–45
Height (m)	1.62 ± 0.08 (1.58–1.65)	1.51–1.85
Weight (kg)	60.37 ± 7.82 (56.71–64.03)	45–80
Body Mass Index (kg/m^2^)	22.98 ± 1.74 (22.17–23.80)	18.7–25

SD: standard deviation; m: meter; kg: kilogram; CI: Confidence Interval.

**Table 2 sensors-24-06884-t002:** Mean and standard deviation of test and retest values of gait kinematic.

LATERAL VIEW			
Right Lower Limb	Test-1 Mean ± SD	Test-2 (Retest) Mean ± SD	Reference Value
**Ankle Angle**			
Initial Contact	99.04 ± 3.19	99.57 ± 3.50	90° to 95°
Loading Response	98.63 ± 4.34	99.18 ± 4.86	90° to 96°
Mid Stance	86.80 ± 5.41	87.30 ± 4.50	78° to 86°
Terminal Stance	84.75 ± 4.22	85.61 ± 3.50	76° to 84°
Pre Swing	108.25 ± 6.15	108.93 ± 5.59	99° to 109°
Initial Swing	99.90 ± 3.28	101.58 ± 3.29	94° to 104°
Mid Swing	94.54 ± 4.37	95.85 ± 3.65	87° to 93°
**Knee Angle**			
Initial Contact	177.47 ± 2.60	178.52 ± 2.17	168° to 178°
Loading Response	161.96 ± 4.22	162.82 ± 3.52	156° to 165°
Mid Stance	170.54 ± 3.03	171.09 ± 2.87	168° to 177°
Terminal Stance	166.45 ± 2.62	167.35 ± 2.50	163° to 171°
Pre Swing	140.12 ± 4.71	140.89 ± 4.56	136° to 147°
Initial Swing	120.58 ± 3.08	121.49 ± 2.63	116° to 126°
Mid Swing	150.56 ± 4.19	151.92 ± 3.89	146° to 157°
**Hip Angle**			
Initial Contact	21.63 ± 2.15	21.83 ± 1.79	(+) 20° to (+) 27°
Loading Response	21.26 ± 3.77	21.55 ± 3.08	(+) 19° to (+) 26°
Mid Stance	−1.11 ± 2.47	−1.13 ± 2.88	0° to (−) 6°
Terminal Stance	−16.97 ± 3.37	−17.60 ± 2.95	(−) 15° to (−) 23°
Pre Swing	−9.36 ± 2.49	−9.90 ± 2.59	(−) 7° to (−) 15°
Initial Swing	17.66 ± 3.67	18.06 ± 3.33	(+) 9° to (+) 17°
Mid Swing	28.54 ± 2.67	29.00± 2.89	(+) 22° to (+) 30°
**POSTERIOR VIEW**			
**Rear Foot Angle**			
Mid stance	5.19 ± 1.64	5.14 ± 1.76	(+) 2° to (+) 6°
Pelvic Drop			
Mid stance	2.65 ± 1.43	3.07 ± 1.61	0° to (+) 5°

SD: standard deviation. All values are presented in degrees.

**Table 3 sensors-24-06884-t003:** Mean and standard deviation of test and retest values of posture parameters.

ANTERIOR VIEW			
Parameters	Test-1 Mean ± SD	Test-2 (Retest) Mean ± SD	Reference Value
Horizontal alignment of the Head	1.04 ± 0.56	0.97 ± 0.5	0^o^
Horizontal alignment of the Acromions	1.33 ± 0.86	1.30 ± 0.90	0^o^
Horizontal alignment of the ASIS’s	1.05 ± 0.72	1.13 ± 0.75	0^o^
Q Angle (Right LL)	13.29 ± 2.62	13.33 ± 2.31	Men: <(+) 15° Women: <(+) 20°
**POSTERIOR VIEW**			
Rear Foot Angle (Right LL)	3.30 ± 1.65	3.23 ± 1.29	(−) 5° to (+) 5°
**RIGHT LATERAL VIEW**			
Forward Head Angle	51.37 ± 2.35	52.18 ± 3.16	>50
Shoulder Angle	53.36 ± 3.20	53.68 ± 2.57	>52°
Genu Recurvatum (Right LL)	2.52 ± 1.52	2.83 ± 1.62	<(−) 10°

SD: standard deviation; Q: quadriceps angle; LL: lower limb. All values are presented in degrees.

**Table 4 sensors-24-06884-t004:** Intra-rater reliability, SEM and SDD of the gait parameters of right lower limb.

LATERAL VIEW				
Right Lower Limb	ICC	95% CI	SEM	SDD
**Ankle Angle**				
Initial Contact	0.913	0.780–0.960	0.99	2.74
Loading Response	0.934	0.833–0.974	1.18	3.28
Mid Stance	0.940	0.848–0.976	1.22	3.38
Terminal Stance	0.950	0.875–0.980	0.87	2.40
Pre Swing	0.952	0.878–0.981	1.29	3.57
Initial Swing	0.90	0.747–0.960	1.04	2.88
Mid Swing	0.915	0.786–0.966	1.17	3.25
**Knee Angle**				
Initial Contact	0.958	0.894–0.983	0.49	1.36
Loading Response	0.965	0.910–0.986	0.73	2.02
Mid Stance	0.929	0.820–0.972	0.79	2.18
Terminal Stance	0.902	0.752–0.961	0.80	2.22
Pre Swing	0.972	0.930–0.989	0.78	2.15
Initial Swing	0.908	0.768–0.964	0.87	2.41
Mid Swing	0.969	0.922–0.988	0.71	1.97
**Hip Angle**				
Initial Contact	0.903	0.754–0.961	0.62	1.71
Loading Response	0.922	0.803–0.969	0.96	2.66
Mid Stance	0.913	0.780–0.966	0.79	2.19
Terminal Stance	0.957	0.893–0.983	0.66	1.82
Pre Swing	0.979	0.947–0.992	0.37	1.02
Initial Swing	0.916	0.788–0.967	1.02	2.82
Mid Swing	0.915	0.784–0.966	0.81	2.25
**POSTERIOR VIEW**				
**Rear Foot Angle**				
Mid Stance	0.904	0.759–0.962	0.53	1.46
Pelvic Drop				
Mid Stance	0.929	0.821–0.972	0.41	1.12

ICC: intraclass correlation coefficient; CI: confidence interval; SEM: standard error of measurement; SDD: smallest detectable difference. All values are presented in degrees.

**Table 5 sensors-24-06884-t005:** Intra-rater reliability, SEM and SDD of the posture parameters.

ANTERIOR VIEW				
	ICC	95% CI	SEM	SDD
Horizontal alignment of the Head	0.925	0.811–0.970	0.15	0.40
Horizontal alignment of the Acromions	0.945	0.861–0.978	0.21	0.57
Horizontal alignment of the ASIS’s	0.943	0.857–0.978	0.18	0.49
Q Angle (Right LL)	0.904	0.757–0.962	0.77	2.12
**POSTERIOR VIEW**				
Rear Foot Angle (Right LL)	0.947	0.865–0.979	0.34	0.95
**RIGHT LATERAL VIEW**				
Forward Head Angle	0.912	0.777–0.965	0.83	2.29
Shoulder Angle	0.902	0.751–0.961	0.91	2.52
Genu Recurvatum (Right LL)	0.925	0.811–0.970	0.43	1.19

ICC: intraclass correlation coefficient; CI: confidence interval; SEM: standard error of measurement; SDD: smallest detectable difference. LL: lower limb. All values are presented in degrees.

**Table 6 sensors-24-06884-t006:** Glossary.

Posture Glossary	
Sign	Indication
+	Inclination to the patient’s right
−	Inclination to the patient’s left
+	Knee valgus
−	Knee varus
+	Rear foot eversion
−	Rear foot inversion
+	Knee flexion
−	Knee hyperextension (Genu recurvatum)
**Walking Glossary**	
**Sign**	**Indication**
>90°	Ankle plantarflexion
<90°	Ankle dorsiflexion
>180°	Knee hyperextension
<180°	Knee flexion
+	Hip flexion
−	Hip extension
+	Rear foot eversion
−	Rear foot inversion
+	Contralateral pelvic drop
−	Ipsilateral pelvic drop
+	Knee Ab/Adduction (Patella is medial to the 2nd toe)
−	Knee Ab/Adduction (Patella is lateral to the 2nd toe)

## Data Availability

The datasets generated and analyzed during this study are accessible from the corresponding author upon reasonable request. For any inquiries related to this paper, please reach out to Saurabh Sharma at ssharma@jmi.ac.in.

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
