# Peer review of "Establishing the Reliability of the GaitON^®^ Motion Analysis System: A Foundational Study for Gait and Posture Analysis in a Healthy Population"

_sensors, 2024, doi:10.3390/s24216884_

Round 1
Reviewer 1 Report
Comments and Suggestions for Authors
In the manuscript, authors study aims to assess reliability of the GaitON system for gait and standing posture analysis. In the study, authors analyze parameters such as ICC, SEM and SDD.
The following are some findings:
I consider that some Figures are not necessary, for example 4(a), 5(c). Instead, some other Figures should be added to explain relevant processes such as the described in section: Precision Angle Measurement and Video Analysis.
It is not completely clear the variables used to measure ICC. That is, if it is a correlation index, what are the pair of variables analyzed through it? The same applies to SEM and SDD. I consider that more detail about the considered variables is necessary and the reasons that led to study these parameters.
In general, there are some Figures that seem redundant. Instead, some Figures more informative about the gait and posture analysis when it was right or wrong in some participants should be included.
Comments on the Quality of English LanguageMinor editing of English language required.
Author Response
Comment: In the manuscript, authors study aims to assess reliability of the GaitON system for gait and standing posture analysis. In the study, authors analyze parameters such as ICC, SEM and SDD. The following are some findings:
I consider that some Figures are not necessary, for example 4(a), 5(c). Instead, some other Figures should be added to explain relevant processes such as the described in section: Precision Angle Measurement and Video Analysis.
Response: We appreciate your valuable feedback and have carefully considered your observations. As this is the first foundational study on GaitOn, we have provided figures to enhance clarity, especially in relation to the equipment used. Upon reviewing your comments, we acknowledge that Figure 4(a) may not be essential to the study, and therefore, we have removed it from the manuscript. However, we would like to emphasize the importance of Figure 5(c). This figure is crucial as it illustrates the camera adjustment process within the gait lab, a key step to ensure accurate capture of gait parameters. Correct calibration of the axes in both the anterior and lateral views is essential, as misalignment could result in erroneous data collection. Furthermore, this figure serves as a practical guide for future researchers using this system, helping them avoid potential calibration errors that could compromise the validity of their findings. Therefore, we believe its inclusion is necessary for both the accuracy of the study and as a reference for the scientific community.
Thank you for your insightful observation regarding the addition of relevant figures. We agree that including figures to explain the processes described in the "Precision Angle Measurement and Video Analysis" section is crucial for clarity. Indeed, we had inadvertently missed this, and we appreciate you bringing it to our attention. In response to your valuable suggestion, we have added new figures (Figures 7a, 7b, 7c and 7d) at the appropriate place in the manuscript. These figures will provide a clearer understanding of the precision angle measurement process and its integration with video analysis.
Comment: It is not completely clear the variables used to measure ICC. That is, if it is a correlation index, what are the pair of variables analyzed through it? The same applies to SEM and SDD. I consider that more detail about the considered variables is necessary and the reasons that led to study these parameters.
Response: Thank you for your valuable feedback.
The variables assessed in this study include kinematic data captured during key phases of the gait cycle, specifically from the lateral view for the right lower limb and from the posterior view. In addition, ICC values were calculated for standing posture parameters, analyzed from the anterior, posterior, and right lateral views. These measurements were carefully chosen to ensure a comprehensive evaluation of both dynamic gait and static postural parameters (Table 2-4). We have now explicitly detailed these variables in the manuscript to enhance clarity regarding the variables used for ICC, SEM, and SDD assessments. The relevant sections have been updated as follows: the aim of the study has been revised in the "Introduction" section, and the first paragraph of the "Results" section has also been modified. These changes have been clearly highlighted for your review.
Comment: In general, there are some Figures that seem redundant. Instead, some Figures more informative about the gait and posture analysis when it was right or wrong in some participants should be included.
Response: Thank you for your valuable feedback. We appreciate your insight regarding the figures. We have removed one figure that was deemed unnecessary for the clarity and focus of the study. Additionally, in response to your suggestion, we have revised the figures to provide more meaningful information related to gait and posture analysis. Specifically, we have added new images that illustrate the angle measurements during walking and standing posture. Essentially, Figures 7a, 7b, 7c and 7d now explain these aspects in greater detail. We believe these additions will enhance the clarity and relevance of the results presented in the study.
Reviewer 2 Report
Comments and Suggestions for Authors
I found the article interesting from the beginning, precisely because I have an osteoarthritis problem caused by my overweight and sedentary lifestyle and I was pleased to learn more about it.
The following are some aspects that I consider important for a better clarity of the paper:
- The introduction should extend the discussion on gait and posture systems both in 2D and 3D, explaining what are the advantages of one or the other beyond cost, it could be in terms of applicability, accuracy, etc. Similarly, I believe it would be useful to know the limitations of both systems, which may require the inclusion of new references to research dealing with these aspects.
- With respect to the participants, I see that initially all are healthy. Could you justify why in the initial study only these people were considered and not with certain health conditions, a reader at this part of the article might ask: if I am getting data from people with good posture and gait why don't I get data from people with unhealthy conditions to test the system?
- Due to the number of participants and the wide age range, the descriptive statistics can be confusing, there are few participants and a table with ages, weights, heights, body mass, in ranges could be presented.
- An analysis of the software used (GaitON) should be provided:
* History of its development
* Comparison with other systems or technologies, why did you choose this one?
* Technical details of the software, hardware, software components, how it processes motion data, necessary calibrations unique to the system, limitations, and real life use cases of the system are needed. It may be necessary to add a subsection dedicated to these details of the software used.
- I recommend that authors include detailed explanations of the statistical metrics used, as well as a justification for their choice and their specific application, which can help readers have a complete understanding of how the data were analyzed.
- I suggest that the authors should highlight that the results of their study will provide a benchmark or starting point for understanding how the GaitON® system can be applied to patients with musculoskeletal conditions and that future research could explore these populations to assess its broader clinical applicability.
I hope my remarks will help improve the clarity of the paper.
Author Response
Comment: I found the article interesting from the beginning, precisely because I have an osteoarthritis problem caused by my overweight and sedentary lifestyle and I was pleased to learn more about it.
Response: Thank you for your kind feedback. We're glad the article resonated with you, especially given your experience with osteoarthritis. We hope the insights provided are helpful in understanding and managing the condition. Thank you again for your thoughtful review.
Comment: The following are some aspects that I consider important for a better clarity of the paper:
- The introduction should extend the discussion on gait and posture systems both in 2D and 3D, explaining what are the advantages of one or the other beyond cost, it could be in terms of applicability, accuracy, etc. Similarly, I believe it would be useful to know the limitations of both systems, which may require the inclusion of new references to research dealing with these aspects.
Response: Thank you for your insightful feedback. In response to your suggestion, we have expanded the introduction to include a more detailed comparison of both 2D and 3D gait and posture analysis systems. This discussion now addresses the advantages of each system beyond cost, focusing on aspects such as applicability and accuracy. Additionally, we have incorporated the limitations of both systems, as suggested, and added relevant references to research that addresses these points. We believe these revisions strengthen the discussion and provide a more comprehensive overview of the topic.
Comment: With respect to the participants, I see that initially all are healthy. Could you justify why in the initial study only these people were considered and not with certain health conditions, a reader at this part of the article might ask: if I am getting data from people with good posture and gait why don't I get data from people with unhealthy conditions to test the system?
Response: Thank you for your thoughtful comment. The current study represents a smaller part of a larger trial, and the data used in this study was drawn from the screening phase, which focused on healthy participants with typical gait and posture. This approach allowed us to establish a solid baseline, ensuring the system’s accuracy in controlled conditions before moving on to more complex cases. However, as part of our next steps, we are extending this research to include participants with specific health conditions, such as pronated feet subjects and subjects having pronated feet with low back pain, to test the system’s robustness and adaptability in more challenging scenarios.
In addition to this, ongoing studies are being conducted on a variety of populations, including those with osteoarthritis of the knee, forward head posture, athletes, and individuals with low back pain. These expanded trials will allow for a more comprehensive evaluation of the system’s performance across diverse conditions, providing deeper insights into its applicability. We appreciate your suggestion and look forward to sharing the results of these broader investigations in future publications.
Comment: Due to the number of participants and the wide age range, the descriptive statistics can be confusing, there are few participants and a table with ages, weights, heights, body mass, in ranges could be presented.
Response: Thank you for your valuable comment. To address your concern regarding the descriptive statistics and the potential for confusion due to the wide age range and number of participants, we have now added detailed information in Table 1. This table includes participants' ages, weights, heights, and body mass, presented in ranges for better clarity and ease of interpretation. Additionally, the relevant changes have been highlighted for your review to ensure clarity. We believe this addition will provide a clearer and more organized overview of the participants' characteristics.
Comment: An analysis of the software used (GaitON) should be provided:
* History of its development
Response: Thank you for your comment. We have briefly mentioned the development and background of the GaitON® system in the manuscript in the introduction section. The relevant sections have been highlighted to provide a clearer understanding of its evolution and application in gait and posture analysis.
Comment: Comparison with other systems or technologies, why did you choose this one?
Response: Thank you for your comment. We have addressed the comparison with other systems and the rationale for choosing GaitON® in the first paragraph of the Discussion section. This section provides a detailed explanation of the system's advantages and why it was selected for our study.
Comment: Technical details of the software, hardware, software components, how it processes motion data, necessary calibrations unique to the system, limitations, and real life use cases of the system are needed. It may be necessary to add a subsection dedicated to these details of the software used.
Response: Thank you for your valuable comment. We have addressed the technical details of the software, hardware, software components, how the GaitON® system processes motion data, and the necessary calibrations unique to the system in the manuscript under their respective sections. Some points, however, were missing and have been included in the revised version.
It is challenging to present all these aspects in a single paragraph, as they are part of different sections within the manuscript. Combining them would lead to unnecessary repetition. Specifically, the software, hardware, software components, motion data processing, and necessary calibrations are explained in the "Methodology" section under various subsections. The limitations and real-life use cases are provided in separate headings towards the end of the manuscript, following the discussion section.
Comment: I recommend that authors include detailed explanations of the statistical metrics used, as well as a justification for their choice and their specific application, which can help readers have a complete understanding of how the data were analyzed.
Response: Thank you for your valuable suggestion. In response, we would like to clarify that in the "Analysis" section (2.7. Statistical Analyses), we have already mentioned the software used for our statistical calculations. Specifically, we employed IBM SPSS Statistics version 27 (IBM SPSS, Chicago, IL, USA) to calculate intraclass correlation coefficients (ICC) for test-retest reliability. The remaining metrics, such as standard error of measurement (SEM) and smallest detectable difference (SDD), were calculated manually.
Comment: I suggest that the authors should highlight that the results of their study will provide a benchmark or starting point for understanding how the GaitON® system can be applied to patients with musculoskeletal conditions and that future research could explore these populations to assess its broader clinical applicability.
Response: Thank you for your insightful suggestion. We fully agree with your point, and we have now incorporated this in the manuscript in the section strength and highlighted. Specifically, we have highlighted that the results of our study provide a key benchmark for understanding how the GaitON® system can be applied to patients with musculoskeletal conditions. We have emphasized that this study serves as a foundational step, with future research focused on exploring diverse patient populations to further assess the broader clinical applicability of the system. We believe this will be an important direction for extending the system’s relevance and utility in clinical practice.
Comment: I hope my remarks will help improve the clarity of the paper.
Response: Thank you very much for your thoughtful and detailed feedback. We sincerely appreciate the time and effort you have taken to provide such thorough remarks. Your suggestions have been invaluable in improving the clarity and overall quality of the manuscript. Each of your comments has helped us refine key aspects of the study, and we believe the revisions made in response to your insights have significantly strengthened the presentation of our work.
Reviewer 3 Report
Comments and Suggestions for Authors
This study aims to evaluate the intra-rater reliability of the GaitON® system for gait and standing posture analysis. Standard error of measurement (SEM) and least detectable difference (SDD) are also examined in these assessments.
The results obtained in this study are promising, but for validation they should be compared with similar results illustrated in the literature.
Comments on the Quality of English LanguageMinor edits are required.
Author Response
Comment: This study aims to evaluate the intra-rater reliability of the GaitON® system for gait and standing posture analysis. Standard error of measurement (SEM) and least detectable difference (SDD) are also examined in these assessments.
Response: Thank you for accurately summarizing the aim of our study. Indeed, this study focuses on evaluating the intra-rater reliability of the GaitON® system for gait and standing posture analysis. In addition, we have examined important statistical metrics such as the Standard Error of Measurement (SEM) and the Least Detectable Difference (SDD) to provide a thorough assessment of the system's reliability.
Comment: The results obtained in this study are promising, but for validation they should be compared with similar results illustrated in the literature.
Response: Thank you for your thoughtful comment. We appreciate your suggestion regarding validation through comparison with similar studies. As this study focuses on evaluating the intra-rater reliability of the GaitON® system, our primary objective was to establish its consistency and reliability in gait and posture analysis. While we recognize the importance of validating the system by comparing it with similar results in the literature, this falls beyond the scope of the current study. However, we did compare our findings with values reported in other relevant studies to provide context and support for our results. Moving forward, we fully intend to explore the system’s validity in future research, where we will compare it against existing standards and similar systems to further assess its clinical applicability.
Reviewer 4 Report
Comments and Suggestions for Authors
The work presents a study aimed at validating the GaitOn® system for these specific purposes. The description of the results and methodology is fairly clear. However, in the opinion of this reviewer, there are two issues that need to be addressed before the paper is ready for publication.
Firstly, it is unclear what the GaitOn®system is being compared against. The proposed performance metrics focus on accuracy and precision, but in relation to what? It is not clear to this reviewer what the reference system is. What gold standard is being used for comparison?
Secondly, another important point that requires further elaboration is the statement that "the system’s low cost and user-friendly interface may make it suitable for use in telemedicine or home-based rehabilitation programs." This claim needs to be substantiated. Specifically, "low-cost" compared to what? As for the "user-friendly interface," how does this align with the fact that an expert rater conducted the setup and the evaluation in your experiment? How could this system be used at home? There are systems in the literature (e.g. https://doi.org/10.1109/ATEE.2013.6563426, https://doi.org/10.1109/RTEICT.2017.8256802, https://doi.org/10.1109/TBCAS.2022.3173586, https://doi.org/10.1109/WCSN.2014.80) that create a body area network not only for monitoring gait in a very controlled environments (e.g., over a 6-meter walk) but also in activities of daily living (ADLs) with sufficient precision and at a very low cost. These systems should be cited for completeness and potential comparison in order to contextualize the performance and affordability of the GaitOn® system.
Minor
1) The authors should check that all the used acronyms are explained and not repeated every time (e.g. ICC, CI, SEM, SDD, etc).
2) Mainly the English is good and there are only a few typos. However, the paper should be carefully rechecked.
3) Please specify the unity of measurement
Comments on the Quality of English LanguageMainly the English is good and there are only a few typos. However, the paper should be carefully rechecked.
Author Response
Comment: The work presents a study aimed at validating the GaitOn® system for these specific purposes. The description of the results and methodology is fairly clear. However, in the opinion of this reviewer, there are two issues that need to be addressed before the paper is ready for publication.
Response: Thank you for your positive feedback regarding the clarity of our methodology and results. We appreciate the opportunity to address this point and further clarify the scope of our study.
It is important to note that this study is not designed as a validation study, but rather as an assessment of the test-retest intra-rater reliability of the GaitON® system. The primary aim of our research was to evaluate the system's consistency and repeatability in measuring gait and posture parameters within a healthy population when used by the same operator over multiple sessions. We focused on establishing the degree to which the GaitON® system provides stable and consistent measurements across repeated trials, which is a critical component of its reliability.
To make this clearer, we have revised the manuscript to emphasize that this is a reliability study. Specifically, we have highlighted the use of test-retest reliability measures, including intraclass correlation coefficients (ICCs), to demonstrate the system's repeatability. This clarification has been incorporated into the "Introduction" and "Methods" sections to ensure that the study’s objectives are fully aligned with its outcomes. We hope this explanation addresses the concern, and we appreciate your thoughtful review and constructive comments.
Comment: Firstly, it is unclear what the GaitOn® system is being compared against. The proposed performance metrics focus on accuracy and precision, but in relation to what? It is not clear to this reviewer what the reference system is. What gold standard is being used for comparison?
Response: Thank you for this insightful comment. We appreciate the opportunity to clarify the nature of the study. As this is a reliability study, there is no need for a gold standard system for comparison. Instead, our focus is on evaluating the test-retest reliability of the GaitON® system by comparing repeated measurements (Test-1 and Test-2) taken by the same operator under consistent conditions. The aim is to assess the system's ability to produce consistent results over multiple trials, which is a critical factor in determining its reliability for gait and posture analysis. To assess this, we utilized intraclass correlation coefficients (ICCs), which are widely accepted as robust measures of test-retest reliability.
Comment: Secondly, another important point that requires further elaboration is the statement that "the system’s low cost and user-friendly interface may make it suitable for use in telemedicine or home-based rehabilitation programs." This claim needs to be substantiated. Specifically, "low-cost" compared to what? As for the "user-friendly interface," how does this align with the fact that an expert rater conducted the setup and the evaluation in your experiment? How could this system be used at home? There are systems in the literature (e.g. https://doi.org/10.1109/ATEE.2013.6563426, https://doi.org/10.1109/RTEICT.2017.8256802, https://doi.org/10.1109/TBCAS.2022.3173586, https://doi.org/10.1109/WCSN.2014.80) that create a body area network not only for monitoring gait in a very controlled environments (e.g., over a 6-meter walk) but also in activities of daily living (ADLs) with sufficient precision and at a very low cost. These systems should be cited for completeness and potential comparison in order to contextualize the performance and affordability of the GaitOn® system.
Response: Thank you for your valuable comment and for raising these important points.
We have clarified the "low cost" claim by providing a more specific comparison. The GaitON® system offers a more affordable alternative to traditional motion capture systems, such as the Vicon®, which are often prohibitively expensive. To provide further clarity, we have included a reference (Reference No- 30) supporting this comparison in the ‘introduction’ section. This makes the GaitON® system a more accessible option for broader use in clinical and rehabilitation settings.
Thank you again for raising another important point. To avoid any confusion, we have removed the statement regarding the system’s suitability for use in telemedicine or home-based rehabilitation programs. While the GaitON® system does have a user-friendly interface, in this study, setup and evaluation were conducted by an expert. We acknowledge that further investigation is required to determine whether the system could be used effectively at home after proper training. Given the potential for confusion, we decided to remove this section for now and will explore this in future studies.
Additionally, we carefully reviewed the suggested papers and determined that most of them are not directly relevant to the focus of our study, which is on gait and posture analysis. While one of the cited studies mentions gait and posture, it does not specifically focus on reliability. Additionally, the full procedure for gait and posture analysis is not thoroughly detailed in that study.
Moreover, we have decided to remove the part of our manuscript referring to the system’s suitability for use in telemedicine or home-based rehabilitation programs to avoid any confusion, as it extends beyond the current scope of our study, which focuses only on the reliability of the GaitON® system. Hence, we have not included these papers in the revised manuscript but have referenced them in this response file for completeness.
- Roşu, M., & Paşca, S. (2013, May). A WBAN-ECG approach for real-time long-term monitoring. In 2013 8TH International symposium on advanced topics in electrical engineering (ATEE) (pp. 1-6). IEEE.
- Jijesh, J. J. (2017, May). Implementation of health monitoring in sensor platform for wireless body area network. In 2017 2nd IEEE International Conference on Recent Trends in Electronics, Information & Communication Technology (RTEICT) (pp. 1265-1270). IEEE.
- Coviello, G., Florio, A., Avitabile, G., Talarico, C., & Wang-Roveda, J. M. (2022). Distributed full synchronized system for global health monitoring based on FLSA. IEEE Transactions on Biomedical Circuits and Systems, 16(4), 600-608.
- Meng, Y. F., Qin, T. F., & Xing, J. (2014, December). Sensor cooperation based on network coding in wireless body area networks. In 2014 International Conference on Wireless Communication and Sensor Network (pp. 358-361). IEEE.
We greatly appreciate your thoughtful feedback, which has helped us enhance the clarity and accuracy of our manuscript.
Minor
Comment: The authors should check that all the used acronyms are explained and not repeated every time (e.g. ICC, CI, SEM, SDD, etc).
Response: Thank you for bringing this to our attention. We have carefully reviewed the manuscript to ensure that all acronyms (e.g., ICC, CI, SEM, SDD) are properly defined upon their first use and are not repeated unnecessarily. This will enhance the readability and consistency of the manuscript.
Comment: Mainly the English is good and there are only a few typos. However, the paper should be carefully rechecked.
Response: Thank you for your positive feedback regarding the quality of the English in the manuscript. We have carefully rechecked the entire paper to correct any remaining typos and ensure consistency in language usage. We appreciate your suggestion and have taken steps to further improve the overall clarity and presentation of the manuscript.
Comment: Please specify the unity of measurement
Response: Thank you for highlighting this important point. We have reviewed the manuscript and ensured that all units of measurement are clearly specified throughout the text, tables, and figures.
Round 2
Reviewer 4 Report
Comments and Suggestions for Authors
The authors have improved the paper by addressing the questions raised. They have also clarified the purpose of this paper more effectively. However, I honestly find the scientific soundness lacking. This is a reliability test of a commercial product, with no new methodologies or concepts being proposed, nor is there a complete assessment (a market analysis is missing). At this point, I do not believe the paper is suitable for this journal.
Comments on the Quality of English LanguageMinor editing of English language required.
